# Effects of Toxic Heavy Metal Salts on Oxidative Quality Deterioration in Ground Pork Model during Aerobic Display Storage

**DOI:** 10.3390/antiox11071310

**Published:** 2022-06-30

**Authors:** Youn-Kyung Ham, Dong-Heon Song, Hyun-Wook Kim

**Affiliations:** 1Department of Animal Science, Sangji University, Wonju 26339, Korea; ykham21@sangji.ac.kr; 2Animal Products Research and Development Division, National Institute of Animal Science, Rural Development Administration, Wanju 55365, Korea; timesoul@naver.com; 3Department of Animal Science & Biotechnology, Gyeongsang National University, Jinju 52725, Korea; 4Department of GreenBio Science, Gyeongsang National University, Jinju 52725, Korea

**Keywords:** aerobic display, discoloration, lipid oxidation, protein oxidation, toxic heavy metal

## Abstract

The contamination of toxic heavy metals in meat production and processing can cause the oxidative deterioration of processed meat products. Aside from the possible mechanisms of toxic heavy metals on pro-oxidative reaction, little is known about the potential impacts of toxic heavy metal contamination on meat quality attributes within permitted maximum residual levels. Therefore, the objective of this study was to determine the influence of the intentional contamination of toxic heavy metals on the oxidative deterioration in ground pork models during aerobic display storage. Four types of toxic heavy metal salts (As_2_O_3_, CdCl_2_, K_2_Cr_2_O_7_, and Pb(NO_3_)_2_) were mixed with ground pork at two different levels (maximum residue limit and its half level), PVC-wrapped, and displayed in a 4 °C showcase equipped with continuous fluorescent natural white light (1400 l×, color temperature = 6500 K). The contamination of toxic heavy metals significantly decreased the redness of ground pork, and rapidly increased the hue angle. The contamination of Cd and Cr equivalent to maximum residue levels (0.05 and 1.0 mg/kg, respectively) could increase the formation of peroxides, 2-thiobarbituric acid reactive substances, and carbonyls, along with an immediate decrease in total reducing activity. However, there was no difference in protein thiol content between treatments (*p* > 0.05). These results indicate that contamination of certain toxic heavy metals, particularly Cd and Cr, would accelerate discoloration, lipid oxidation, and carbonyl formation of ground pork during aerobic storage.

## 1. Introduction

The risk of chemical contamination in the meat production system has increased due to rapid industrialization and urbanization [1]. The ingestion of meat contaminated with toxic heavy metals causes bioaccumulation in the human body, which potentially leads to central nervous system disorders in severe cases. Toxic heavy metals detected in meat products, including As, Cd, Cr, Pb, and Ni, are generally absorbed in livestock animals from drinking water or feedstuffs [2]. Thus, the types and concentrations of toxic heavy metals detected in market meat products are greatly affected by production region and environment; in this regard, numerous previous studies have focused on determining the cause of the detection of meat contaminated with toxic heavy metals in a specific area [1,3,4]. Despite the seriousness and specificity of toxic heavy metal contamination, however, only a few studies regarding the practical quality change of meat contaminated with toxic heavy metals have been conducted.

Compelling clues to possible quality defects in meat contaminated with toxic heavy metals can be found at the quality attributes of wild and game meat, even if biological information in the previous studies was occasionally unclear and the sample size was too small. For example, some previous reports have found a high level of malondialdehyde (MDA) in wild meat, indicating accelerated lipid oxidation [5,6,7]. Cifuni, et al. [5] noted that MDA levels of fallow deer and wild boar with selective hunting were approximately 3.58 and 3.90 mg/kg meat, respectively. Šuran, et al. [7] reported a high correlation between MDA and Pb concentration in young wild boars (1 to 3 years old). Therefore, it could be expected that meat contaminated with toxic heavy metals may be vulnerable to the oxidation of biological molecules, such as protein and lipid, resulting in oxidative quality defects.

In general, oxidative quality deterioration of meat products includes discoloration, toughness, off-flavor, and nutritional loss, which mainly result from the oxidation of lipid and protein molecules [8]. Lipid autoxidation, which is triggered by the reaction of unsaturated fatty acid with oxygen molecules, forms primary oxidative products, such as hydroperoxide, and this unstable substance can be decomposed into peroxy and alkoxy radicals by catalysis of mainly iron in heme-proteins and free iron ions [9]. This reaction is well known as the Fenton reaction, and similarly, the Fenton-like reaction can occur with transition metals, such as Cu^2+^, Ti^4+^, and Co^3+^ [9,10]. Moreover, free radicals and primary/secondary oxidative products produced during lipid oxidation can modify the protein backbone or the side chain of amino acids, which initiate protein oxidation [11].

Therefore, our experimental hypothesis could be that contamination of toxic heavy metals, particularly Cr and Cd included in the transition metal group, may accelerate lipid and protein oxidation of meat products during storage. Moreover, the results of this current study may provide a new insight to discuss the quality defects of meat products caused by contamination of toxic heavy metals. Since the level of toxic heavy metal contamination in an alive biological sample cannot be precisely controlled, in this study, color change, lipid oxidation, and protein oxidation in ground pork model intentionally contaminated with toxic heavy metal salts were evaluated during 7 days of aerobic display storage.

## 2. Materials and Methods

### 2.1. Experimental Design and Arrangement

An experimental arrangement was a 9 (treatment) × 3 (storage period) factorial set in a completely randomized block design with three independent batches (*n* = 3), where treatment effect, storage effect, and their interaction effects were fixed as main effects. The treatment group included a control without heavy metal salt addition and treatments intentionally contaminated with four types of toxic heavy metals (As, Cd, Cr, and Pb) with two different levels, as toxic heavy metals detected in meat products with relatively high frequency [2]. Two concentrations (maximum residue limit and its half level) of each toxic heavy metal were set considering the maximum residue limit by the USDA Foreign Agricultural Service (USDA FAS) and the EFSA (Table 1); limit of total As in meat, 0.05 mg/kg [12]; maximum level of Cd in meat (excluding offal) of bovine animals, sheep, pig, and poultry, 0.050 mg/kg wet weight [13]; limit of Cr in meat (including liver and kidney), 1.0 mg/kg [12]); maximum levels of Pb in meat (excluding offal) of bovine animals, sheep, pig, and poultry, 0.10 mg/kg wet weight [13].

### 2.2. Raw Materials and Chemicals

Fresh pork hams (*Musculus biceps femoris*, *M. adductor*, *M. semitendinosus*, and *M. semimembranosus*) were purchased from a local processor (Jinsan butcher shop, Jinju, Korea) at post-mortem 48 h. Arsenic (III) oxide (As_2_O_3_), cadmium chloride (CdCl_2_), potassium dichromate (K_2_Cr_2_O_7_), and lead (II) nitrate (Pb(NO_3_)_2_) were purchased from Sigma-Aldrich (St. Louis, MO, USA), and all other reagents used were of analytical grade.

### 2.3. Sample Preparation and Simulated Aerobic Display Condition

Toxic heavy metal stock solutions were prepared at 10 times the target concentrations to be residue in the ground pork, in consideration of the molar mass of toxic heavy metals in each dissolved salts; As (74.9216 g/mol in 197.841 g/mol of As_2_O_3_), Cd (112.411 g/mol in 183.32 g/mol of CdCl_2_), Cr (51.9961 g/mol in 294.185 g/mol of K_2_Cr_2_O_7_), and Pb (207.2 g/mol in 331.2 g/mol of Pb(NO_3_)_2_). All weighed toxic heavy metal salts were dissolved in distilled deionized water (DDW) and stored in a 4 °C refrigerator until used.

Excessive subcutaneous fat and visible connective tissue of fresh pork hams were removed, and the trimmed pork hams were cut into approximately 2 × 2 cm^2^ block size. The meat blocks (approximately 18 kg from four pork hams per batch) were ground through an 8 mm plate using a meat grinder, and the ground pork was separated into nine portions (1.8 kg each). The remaining portion of ground pork was used to evaluate the initial contamination levels of toxic heavy metals according to Ham, et al. [14]. The toxic heavy metal content was measured using the sequential procedure of microwave digestion and inductively coupled plasma optical emission spectrometry (ICP-OES, Perkin Elmer Optima 5300DV, Perkin Elmer, Llantrisant, UK) according to the manufacturer’s manual (Application Note UW-19). The sample was mixed with concentrated nitric acid and wet-digested in a microwave oven. Toxic heavy metals (As, Cd, Cr, and Pb) in the digested and diluted samples and black solutions were determined using the ICP-OES system. Recovery experiments were conducted using three different concentrations of known spiked samples. Concentrations of toxic heavy metals below the limit of detection were expressed as “less than detectable limit”. The fat content of ground pork was determined as 8.53 ± 0.23 g/100 g according to the Soxhlet method.

In each treatment, one portion of the ground pork (1.8 kg) was mixed with 200 mL of each prepared toxic heavy metal stock solution as mentioned above, and a control portion was prepared with only 200 mL of DDW. The ground pork intentionally contaminated with toxic heavy metals was separated into three sub-portions (600 g each), and each portion was individually placed on an extended polystyrene (Styrofoam) tray, wrapped with commercial polyvinyl chloride (PVC) film, and displayed in a 4 °C showcase equipped with continuous fluorescent natural white light (1400 l×, color temperature = 6500 K). The displayed samples were used for pH, instrumental color, total reducing activity, peroxide value, 2-thiobarbituric acid reactive substances, protein carbonyl content, and protein thiol content at 1 (Day 1), 4 (Day 4), and 7 days (Day 7) of the simulated aerobic display storage. A total of three batches was repeated on each different day.

### 2.4. Analysis of Ground Pork with Toxic Heavy Metals

#### 2.4.1. pH Value

The pH value of ground pork homogenates (ten-fold diluted) was measured in triplicate using an electric pH meter (Orion star A211; Thermo Fisher Scientific, Waltham, MA, USA). The pH meter was calibrated at room temperature using pH standard buffers (pH 4.01, 7.00, and 10.01). Data were reported as the average value of technical triplicates.

#### 2.4.2. Instrumental Color

The surface color of the ground pork was measured using a colorimeter (Chroma meter, CR 400; Minolta, Osaka, Japan) equipped with an 8 mm diameter of aperture. Since the sample was stored under aerobic conditions, the color measurement was performed immediately without blooming time. The setting for the illuminant was D_65_ source, and the observer was standard 2°. The colorimeter was calibrated with a white calibration tile (L* = + 97.83, a* = –0.43, b* = + 1.98) based on the manufacturer’s instructions. The CIE L* (lightness), a* (redness), and b* (yellowness) values were recorded from six random locations and used to calculate Hue angle and total color difference (∆*E*) [15].

#### 2.4.3. Total Reducing Activity

Total reducing activity was determined in triplicate using the method of Lee, Cassens, and Fennema [16]. The total reducing activity (unitless) of ground pork homogenates was expressed as the difference between the absorbance of 1 mM potassium ferricyanide at 420 nm and the absorbance of the sample at 420 nm. Data were reported as the average value of technical duplicates.

#### 2.4.4. Peroxide Value (POV)

POV of ground pork was determined following the method of the International Dairy Federation (IDF) described by Kim, et al. [17]. The lipid fraction (150 mg) extracted by the Soxhlet method was placed in a glass tube and mixed with 9.8 mL of the chloroform/methanol solvent mixture. After vortex-mixing for 5 s, 50 μL of ammonium thiocyanate solution (30 g/100 mL) was added to the glass tube and vortex-mixed again for 5 s. Then, 50 μL of iron (II) solution was immediately added [18], vortex-mixed for 5 s, and incubated at room temperature for 5 min. The absorbance of the mixture was read at 500 nm using a spectrophotometer, and POV was expressed as the average of technical duplicates in milliequivalents of peroxide per kg fat (meq O_2_/kg fat).

#### 2.4.5. Thiobarbituric Acid Reactive Substances (TBARS)

TBARS value of ground pork was determined according to the modified method of Buege and Aust [19] described by Kim, et al. [17]. The TBARS value was calculated using a molecular extinction coefficient (1.56 × 10^5^ M^−1^ cm^−1^) and expressed as the average of technical duplicates in mg malondialdehyde per kg of meat (mg MDA/kg meat).

#### 2.4.6. Protein Carbonyl Content

Protein carbonyl content of ground pork was determined according to the protein carbonyl colorimetric method using 2,4-dinitrophenylhydrazine (DNPH) described by Soyer, et al. [20]. The carbonyl content was expressed as the average of technical duplicates in nmol carbonyls per mg protein (nmol/mg protein).

#### 2.4.7. Thiol Content

Thiol content (sulfhydryl group) of ground pork was determined according to the colorimetric method of Srinivasan and Xiong [21] using 5,5-dithiol-bis-(2-nitrobenzoic acid) (DTNB) reagent. The thiol content was calculated using a molecular extinction coefficient (11,400 M^−1^ cm^−1^) and expressed as the average of technical duplicates in nmol thiol groups per mg protein (nmol/mg protein).

### 2.5. Statistical Analysis

Data from all measured variables were analyzed using the general linear model (GLM) procedure in SPSS 18.0 program (SPSS Inc., Chicago, IL, USA). All trait models included treatment, display period, and treatment × display period interaction as fixed effects and independent batch as a random effect. In the variables with statistically significant effect at 5% critical value, Duncan’s multiple range test was used to determine the significance of the differences between treatments at 5% critical value. However, the post-hoc test was not conducted when the data changes were consistent during display period. All data were expressed as mean ± standard error (S.E).

## 3. Results

In terms of the initial contamination levels of toxic heavy metals in ground pork (Table 1), 15.19 μg/kg of Cr was detected, but As, Cd, and Pb were less than detectable levels. The significance of *p* value from the two-way ANOVA for treatment, storage, and their interaction effects on measured variables is shown in Table 2. Excluding protein carbonyl content, no significant interactions between treatment and storage effects were found.

### 3.1. Changes in pH and Color during Aerobic Display Storage

Changes in pH and color of ground pork intentionally contaminated with toxic heavy metals during 7 days of aerobic display storage are shown in Table 3. The pH value of all treatments ranged from 5.82 to 5.96, in which the pH value of toxic heavy metal treatments showed a decreasing tendency as compared to that of the control (*p* > 0.05). The pH value of ground pork was affected by display storage (*p* < 0.05), but the pH difference between display periods was too small within 0.1 units.

The intentional contamination of toxic heavy metals significantly affected CIE a* (redness) of ground pork during aerobic display storage, but there were no significant differences in CIE L* (lightness) and CIE b* (yellowness) between treatments. Excluding As 0.25 mg/kg treatment, the contamination of toxic heavy metals caused a significantly lower redness of ground pork than control. As a result, the hue angle (an indicator of discoloration) of toxic heavy metal treatments was higher than that of the control (*p* < 0.05). During aerobic display storage, typical patterns of meat discoloration related to a decrease in redness and an increase in the hue angle were observed. The total color difference of ground pork was not affected by the toxic heavy metal contamination, the aerobic display storage, and their interaction (*p* > 0.05). Thus, our results show that contamination of As, Cd, Cr, and Pb within maximum residue levels may accelerate the discoloration of ground pork under aerobic storage condition.

### 3.2. Change in Total Reducing Activity during Aerobic Display Storage

The total reducing activity of ground pork intentionally contaminated with toxic heavy metals during 7 days of aerobic display storage was significantly affected by each treatment and storage period effect (Figure 1). On Day 1, the ground pork intentionally contaminated with As 0.5, Cd 0.025 and 0.05 mg/kg, and Cr 0.5 and 1.0 mg/kg decreased total reducing activity when compared to the control (*p* < 0.05). As the display period passed, the total reducing activity of all treatments was gradually decreased (*p* < 0.05). In addition, although only two treatments (As 0.5 and Cr 0.5 mg/kg) still presented a lower total reducing activity on Day 4 than the control, there was no significant difference in total reducing activity between treatments on Day 7. The results of this study show that the intentional inclusion of specific toxic heavy metals, such as As, Cd, and Cr, could immediately have negative impacts on the initial total reducing activity of ground pork.

### 3.3. Lipid Oxidation during Aerobic Display Storage

The lipid oxidation of ground pork intentionally contaminated with toxic heavy metals was determined through the quantitative evaluation of peroxides (Figure 2) and malondialdehyde formation (Figure 3). On Day 1, all treatments showed significantly higher peroxide values than the control. On Day 4, the peroxide value of all treatments significantly increased. In particular, 3.09 and 3.24 fold greater amounts of peroxide formation in Cd 0.025 and 0.05 mg/kg treatments were observed, respectively, when compared to the control (1.58 meq O_2_/kg). However, on Day 7, the peroxide values of all treatments decreased back to the peroxide levels on Day 1 (*p* > 0.05), and there was no significant difference between the treatments.

No difference in the TBARS values of ground pork samples was observed on Day 1, but from Day 4, the treatment groups showed significantly higher TBARS values than the control (0.04 mg MDA/kg meat). In particular, in ground pork intentionally contaminated with Cd and Cr, a dose-dependent increase in TBARS value was observed on Day 7. Thus, the results of this study show that the formation of primary oxidation products could be immediately accelerated due to toxic heavy metal contamination within the maximum residue limit, which could cause the promotion of secondary oxidation products during aerobic storage.

### 3.4. Protein Oxidation during Aerobic Display Storage

Protein oxidation of ground pork intentionally contaminated with toxic heavy metals was determined through the evaluation of protein carbonyl content (Figure 4) and total thiol content (Figure 5). On Day 1, protein carbonyl content was similar between control and treatments (*p* > 0.05). The carbonyl content of ground pork contaminated with heavy metals increased markedly from Day 4 (*p* < 0.05), and significantly greater formation of the protein carbonyl group was observed in ground pork contaminated with Cd 0.05 mg/kg and Cr 1.0 mg/kg compared to control. Thiol content of ground pork gradually decreased during 7 days of aerobic display storage (*p* < 0.05), but there was no significant difference between control and treatments on each storage day.

## 4. Discussion

Toxic heavy metals, such as As, Ag, Cd, Co, Cr, Hg, Pb, Sn, and Sb, can cause central nervous system disorders through continuous bioaccumulation even if a trace amount is ingested [2]. A recent report that the level of certain toxic heavy metals in human blood differs between vegetarians and non-vegetarians implies that meat products can be a critical source of the bioaccumulation of toxic heavy metals in the human body [22,23]. In practice, detectable levels of As, Cd, Cr, Hg, Ni, and Pb contamination have been found in the meat and muscle tissue of terrestrial animals, including bovine, pigs, and poultry [1,3,4,24]. Moreover, metal contamination on food can occur from food equipment and packaging materials used for harvesting, processing, storage, and cooking [2]. In this regard, related previous studies have mainly focused on evaluating the status of heavy metal contamination of meat in a certain region or at each manufacturing step and studying the effects of heavy metal contaminated meat ingestion on the human body [1,2,3,4].

Recognizable quality deterioration in meat products through the senses of sight, smell, and taste (e.g., surface slime, off-flavor/odor, and discoloration) is a critical indicator that makes consumers’ ingestion suspicious. However, there has been little to no previous literature on the recognizable deterioration in meat contaminated with toxic heavy metals. Nevertheless, there was a previous observation that divalent heavy metals as a pro-oxidant, including Fe^2+^, Cu^2+^, Co^2+^, could promote lipid and protein oxidation of meat products, and a compelling clue providing that oxidative quality degradation may occur with toxic heavy metal contamination [25]. Concerning this hypothesis, our results show that the intentional contamination of As, Cd, Cr, and Pb could lead to discoloration (Table 2), lipid oxidation (Figure 2 and Figure 3), and protein oxidation (Figure 4) of ground pork during 7 days of aerobic display storage. Considering the obtained results, it could be reasonable to understand that the contamination of As, Cd, Cr, and Pb, even within permissible residue limits, can show perceptible quality defects of meat products related to lipid and protein oxidation.

Lipid oxidation is one of the major causes of oxidative quality deterioration, resulting in off-flavor and nutritional loss of meat products during storage. In general, lipid autoxidation proceeds through initiation, propagation, and termination stages, and free radicals and oxidation products produced in the stages can initiate the oxidation of protein and nucleic acid [8]. In this study, the accelerated lipid oxidation of ground pork due to toxic heavy metal contamination was investigated by evaluating the formation of primary (peroxides) and secondary (malondialdehyde) oxidation products. The progress of lipid oxidation showed a typical pattern in which secondary oxidation production increased after reaching the maximum formation of primary oxidation products on Day 4. Therefore, the highest TBARS values of ground pork contaminated with Cd and Cr on Day 7 could imply that lipid oxidation could be accelerated due to contamination of the toxic heavy metals.

Heavy metal ions, such as Fe^2+^, Cu^2+^, Co^2+^, and Ti^4+^, are well known to facilitate lipid oxidation of meat products [9,10,25]. One of the related chemical reactions is called the Fenton reaction, which decomposes hydrogen peroxide by metallic catalysis, and the catalytic reaction by iron ions has been studied mainly in lipid oxidation of meat products; Fe^2+^ + H_2_O_2_ → Fe^3+^ + HO·+ OH^−^ [25]. Moreover, although membrane permeability of Cr ions differs depending on the oxidation state, intracellular conversion of Cr^6+^ to Cr^3+^ can contribute to the generation of hydrogen peroxide, hydroxyl radicals, and superoxide anion radicals possibly associated with the Fenton reaction [26]. Liu, Qu, and Kadiiska [27] have indicated that Cd can produce free radicals, such as superoxide anion, hydroxyl radical, and lipid radicals, due to the indirect involvement of the Fenton reaction. In this study, initially decreased total reducing activity (Figure 1) of ground pork intentionally contaminated with Cd and Cr might be due to the formation of reactive oxidation species possibly related to the Fenton reaction.

The functional group of primary and secondary lipid oxidation products, such as hydroperoxide, peroxide, aldehyde oxidizes protein and amino acid groups, is called protein oxidation, and this reaction includes protein carbonyl formation and the loss of sulfhydryl (thiol) group [11]. In this study, the greater formation of protein carbonyls was found at ground pork contaminated with Cd 0.05 and Cr 1.0 mg/kg compared to the control (Figure 4). This result could be related to the highest TBARS values in the treatment groups (Figure 3). As similar results, Aflanie [28] found that the exposure of blood to As (0.001 and 0.02 mg/L) and Cd (0.003 and 0.006 mg/L) increased MDA levels, as well as advancing oxidation protein products (AOPP) in vitro and suggested high correlations (regression coefficient) between MDA and AOPP in AS and Cd contaminations were 0.87 and 0.81, respectively. Changes in protein carbonyl and total thiol contents as commonly used indicators for protein oxidation of meat products [29]. Many previous studies have reported that protein carbonyl formation and the conversion of thiols to disulfides in meat products with accelerated lipid oxidation during storage [30,31]. In this regard, our results were also in agreement with the results of previous studies because a large amount of carbonyl formation was observed in the treatment group with increased TBARS value. However, no difference in protein thiol content between treatments was observed throughout overall aerobic display storage (Figure 5, *p* > 0.05). Toxic heavy metal ions can react with thiol groups of proteins [32], and DTNB, a key color-developing reagent for measuring thiol group content, can also bind heavy metal ions. Therefore, considering this point, it cannot be overlooked that a relatively lower amount of thiol groups than the actual thiol groups with the DTNB reagent reacted.

Despite the convincing experimental results of this study, however, there are still limitations on definitive conclusions; firstly, the bioaccumulation of toxic heavy metals occurs in living animals, and the exposure type (acute and chronic) and contamination level of toxic heavy metals may cause different physiological responses, potentially indicating the diverse oxidation stability of meat produced from the contaminated livestock. Secondly, we cannot completely overlook the effect of the anions in the toxic heavy metals used in this study on the oxidation stability of meat products, even in small amounts. Lastly, considering the cell membrane permeability, excretion, and accumulation location of toxic heavy metals (mainly organ and adipose tissue), it is still unclear whether they can be involved in the redox reaction in muscle cells or tissues. Nevertheless, we believe that the results of this study—that meat contaminated with toxic heavy metals clearly presented rapid discoloration, lipid oxidation, and carbonyl formation under experimentally controlled conditions—indicate the possibility of oxidative quality defect of meat products caused by toxic heavy metals.

## 5. Conclusions

This study shows that intentional contamination of As, Cd, Cr, and Pb within the residual limits regulated by the USDA Foreign Service and the EFSA could reduce oxidative stability of ground pork, related to discoloration, lipid oxidation, and protein oxidation. In particular, the contamination of the Cd and Cr equivalent to maximum residue levels (0.05 and 1.0 mg/kg, respectively) remarkably caused discoloration and accelerated oxidation of lipid and protein in ground pork during the simulated aerobic display storage. In this regard, it would be best to understand that the obtained results were probably associated with the generation of reactive oxygen species by the involvement of Cd and Cr related to the Fenton reaction. Therefore, we suggest that unintentional contamination during production, processing, and distribution would potentially cause the fatal oxidative quality deterioration of meat products.

## Figures and Tables

**Figure 1 antioxidants-11-01310-f001:**
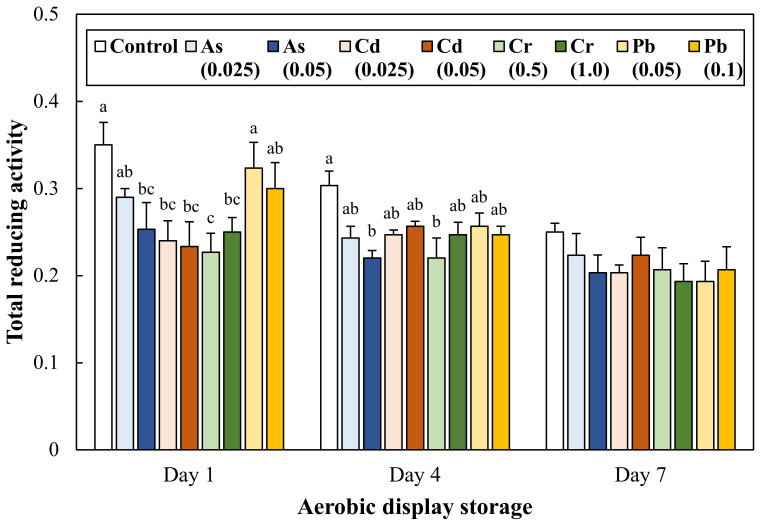
Change in total reducing activity of ground pork intentionally contaminated with toxic heavy metals during 7 days of aerobic display storage. High number of total reducing activity indicates a high ability to reduce ferric iron to ferrous iron. The number in parentheses in legend indicates the target concentration (mg/kg) of each heavy metal in ground pork. Error bars represent each standard error (S.E.). The letters a–c represent results that are not significantly different from each other within each display storage (*p* ≥ 0.05).

**Figure 2 antioxidants-11-01310-f002:**
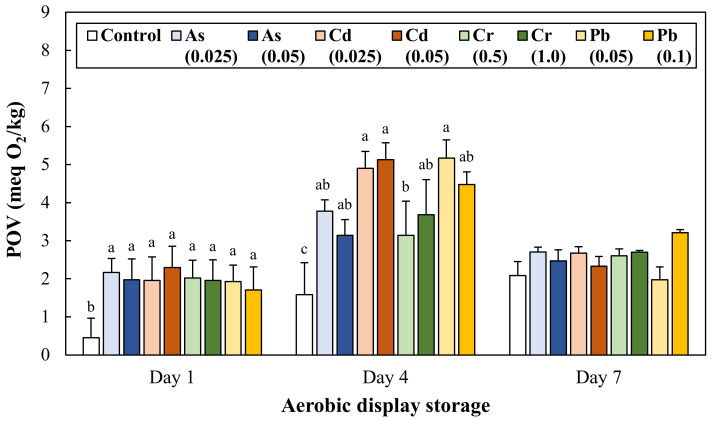
Change in peroxide value (POV) of ground pork intentionally contaminated with toxic heavy metals during 7 days of aerobic display storage. The number in parentheses in legend indicates the target concentration (mg/kg) of each heavy metal in ground pork. Error bars represent each standard error (S.E.). The letters a–c represent results that are not significantly different from each other within each display storage (*p* ≥ 0.05).

**Figure 3 antioxidants-11-01310-f003:**
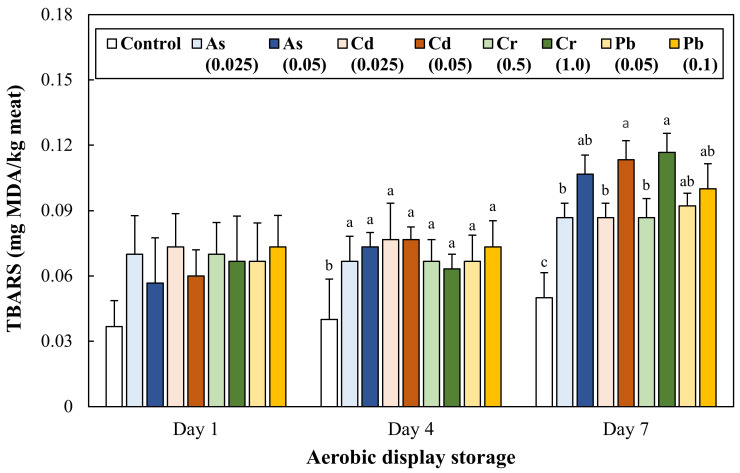
Change in 2-thiobarbituric acid reactive substances (TBARS) of ground pork intentionally contaminated with toxic heavy metals during 7 days of aerobic display storage. The number in parentheses in legend indicates the target concentration (mg/kg) of each heavy metal in ground pork. Error bars represent each standard error (S.E.). The letters a–c represent results that are not significantly different from each other within each display storage (*p* ≥ 0.05).

**Figure 4 antioxidants-11-01310-f004:**
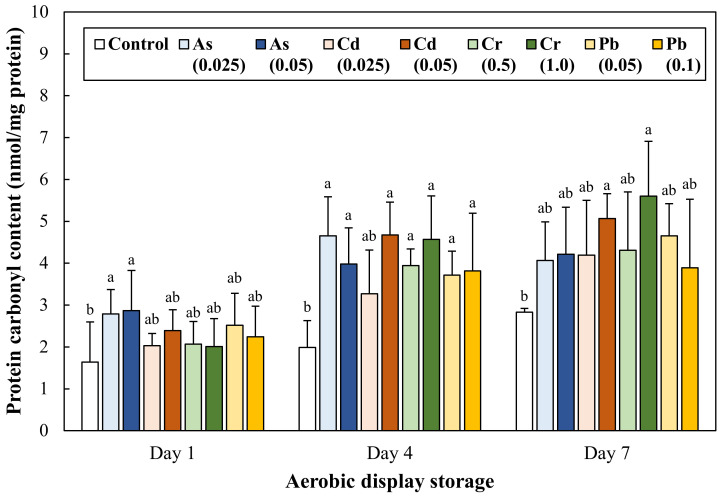
Change in protein carbonyl content of ground pork intentionally contaminated with toxic heavy metals during 7 days of aerobic display storage. The number in parentheses in legend indicates the target concentration (mg/kg) of each heavy metal in ground pork. Error bars represent each standard error (S.E.). The letters a, b represent results that are not significantly different from each other within each display storage (*p* ≥ 0.05).

**Figure 5 antioxidants-11-01310-f005:**
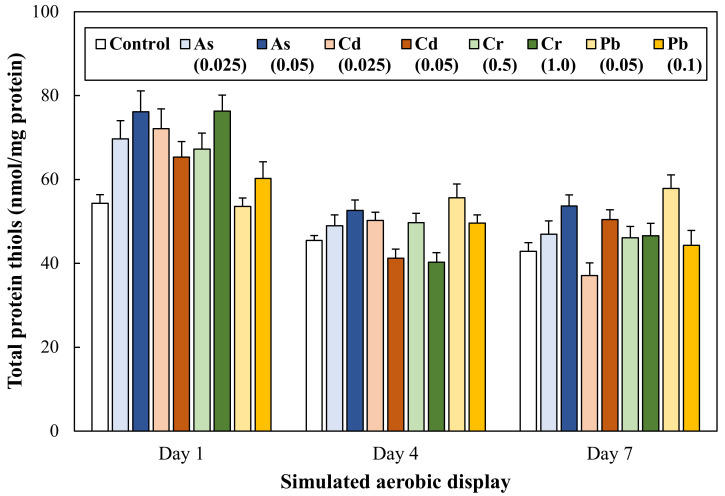
Change in total protein thiol content of ground pork intentionally contaminated with toxic heavy metals during 7 days of aerobic display storage. The number in parentheses in legend indicates the target concentration (mg/kg) of each heavy metal in ground pork. Error bars represent each standard error (S.E.).

**Table 1 antioxidants-11-01310-t001:** Toxic heavy metal salts used and the target concentrations of toxic heavy metals for intentional contamination of ground pork.

Heavy Metal	Used Salt Type	Target Concentration (mg/kg)	Initial Concentration in Ground Pork (μg/kg)
Arsenic (As)	As_2_O_3_	0.025 and 0.05	- ^(1)^
Cadmium (Cd)	CdCl_2_	0.025 and 0.05	-
Chromium (Cr)	K_2_Cr_2_O_7_	0.5 and 1.0	15.19 ± 7.46
Lead (Pb)	Pb(NO_3_)_2_	0.05 and 0.10	-

^(1)^ Less than detectable limit.

**Table 2 antioxidants-11-01310-t002:** The significance of *p* value from the two-way ANOVA for treatment, storage, and their interaction effects on measured variables.

Measured Variables	Treatment Effect (T)	Storage Effect (S)	Interaction (T × S)
pH value	NS ^(1)^	0.035	NS
CIE L* (lightness)	NS	NS	NS
CIE a* (redness)	0.001	0.035	NS
CIE b* (yellowness)	NS	<0.001	NS
Total color difference (∆*E*)	NS	NS	NS
Total reducing activity	0.001	<0.001	NS
Hue angle (discoloration)	<0.001	<0.001	NS
Peroxide value	0.033	<0.001	NS
TBARS	<0.001	<0.001	NS
Carbonyls	0.004	<0.001	0.016
Thiols	NS	0.031	NS

^(1)^ NS: non-significance (*p* ≥ 0.05).

**Table 3 antioxidants-11-01310-t003:** pH and instrumental color changes in ground pork intentionally contaminated with toxic heavy metals during 7 days of aerobic display storage.

**Effect**	**pH**	**Instrumental Color Characteristic**				
		**CIE L*** **(Lightness)**	**CIE a*** **(Redness)**	**CIE b*** **(Yellowness)**	**Hue Angle** **(Discoloration)**	**Total Color Difference (∆*E*)**
*Treatment effect (T)*						
Control	5.96 ± 0.03	53.24 ± 0.57	13.37 ± 0.32a	10.96 ± 0.56	39.18 ± 1.69b	3.74 ± 0.35
As 0.025 mg/kg	5.82 ± 0.04	53.93 ± 0.44	12.84 ± 0.35ab	11.41 ± 0.60	41.56 ± 1.47a	3.19 ± 0.17
As 0.05 mg/kg	5.84 ± 0.03	53.80 ± 0.52	12.09 ± 0.31c	11.42 ± 0.46	43.11 ± 1.21a	3.92 ± 0.51
Cd 0.025 mg/kg	5.84 ± 0.04	52.97 ± 0.39	12.59 ± 0.25bc	11.13 ± 0.49	41.32 ± 1.29a	3.05 ± 0.22
Cd 0.05 mg/kg	5.87 ± 0.04	54.03 ± 0.23	12.44 ± 0.31bc	11.34 ± 0.47	42.21 ± 1.37a	3.08 ± 0.30
Cr 0.5 mg/kg	5.88 ± 0.04	54.86 ± 0.73	12.30 ± 0.44bc	11.13 ± 0.45	42.79 ± 1.92a	3.92 ± 0.45
Cr 1.0 mg/kg	5.85 ± 0.04	53.91 ± 0.86	12.02 ± 0.42c	11.35 ± 0.35	42.58 ± 1.17a	2.81 ± 0.22
Pb 0.05 mg/kg	5.82 ± 0.04	54.11 ± 0.34	12.34 ± 0.28bc	11.48 ± 0.67	42.73 ± 1.31a	2.95 ± 0.48
Pb 0.1 mg/kg	5.85 ± 0.04	54.66 ± 0.59	12.53 ± 0.23bc	11.54 ± 0.50	42.27 ± 1.52a	3.93 ± 0.53
	**pH**	**CIE L*** **(Lightness)**	**CIE a*** **(Redness)**	**CIE b*** **(Yellowness)**	**Hue Angle** **(Discoloration)**	**Total Color Difference (∆*E*)**
*Storage effect (S)*						
Day 1	5.87 ± 0.01xy	54.49 ± 0.36	12.76 ± 0.17x	9.63 ± 0.13z	37.06 ± 0.58z	-
Day 4	5.82 ± 0.01y	53.61 ± 0.24	12.44 ± 0.61xy	11.90 ± 0.13y	43.68 ± 0.44y	3.22 ± 0.21
Day 7	5.89 ± 0.02x	53.74 ± 0.23	12.31 ± 0.12y	12.40 ± 0.13x	45.17 ± 0.43x	3.58 ± 0.15

Mean ± standard error (S.E.); a–c means sharing the same letters within a column among treatments are not significantly different at *p* < 0.05 by Duncan’s multiple range test; x–z means sharing the same letters within a column among storage periods are not significantly different at *p* < 0.05 by Duncan’s multiple range test.

## Data Availability

The data presented in this study are available in the article.

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
