# Peer review of "Effects of Toxic Heavy Metal Salts on Oxidative Quality Deterioration in Ground Pork Model during Aerobic Display Storage"

_antioxidants, 2022, doi:10.3390/antiox11071310_

Round 1
Reviewer 1 Report
The article presents a low scientific level, because the scope of the research does not show any novelty. The effect of heavy metals in the meat as pro-oxidants is well known. In addition, despite the good organization, the article does not present an in-depth analysis of the results obtained, which would allow for exploring the mechanisms, e.g. in the context of the correlation of protein and fat oxidation in meat. The authors used four types of toxic heavy metal salts at two different levels (maximum residue limit and its half level). In the discussion of the results, however, they rarely refer to the two levels used. The authors write about simulated aerobic display condition, however, do not describe how they were obtained? (line 95).
Moreover, the authors conclude by presenting statements that are not confirmed by the presented research results. The hue angle of all the samples to which metal salts were added in two levels was characterized by a significantly higher value compared to the control sample (according to Table 3).
Author Response
Please find the attached file regarding the response to the reviewer's comments. Thank you for your comments and consideration.

Reviewer 2 Report
Relevance of the topic and study design:
This investigation is relevant, and well-conceived. The study design was adequate in general and included the evaluation of meat parameters, both chemical and physical, that impact product quality and safety. L
Title, abstract and keywords:
The title is adequate to the study described on the manuscript.
The abstract is globally suitable, with some of the most relevant findings presented. However, the abstract lacks a small introduction (1-2 sentences) that help readers understand the justification why the study was performed.
The keywords are adequate in significance, and in number also.
Introduction:
The introduction correctly helps to frame and contextualize the work, and includes a number of citations from related works, that allows to see and contextualize the present research, as the objectives indicated in the end of the introduction depict.
Materials and methods:
The description of the experimental techniques is globally acceptable. They allow replication of the study, which is positive. However, I suggest that the authors would calculate the colour difference between all samples and the control. By this, it will possible to evaluate in a more convenient way the real differences induced in the meat.
Please see the following articles for the formula to calculate TDC or DE (both designations are possible) and to see interpretation limits: https://www.mdpi.com/2304-8158/11/7/917
Results and discussion:
In Table 3 a new column should be added with the values of colour difference and these should be included in the discussion.
All tables and Figures were appropriately selected to display the results, and these were sub mitted to statistical analysis, which is beneficial to ensure confidence in the conclusions.
However, Tables and Figures must be fully self-explanatory, and therefore appart from the p-value, you must also add which statistical test was used.
Level of English:
The level of English is acceptable.
Overall comments:
The work is valuable and in essence could be published. However, before that the manuscript needs a minor revision for some additional improvement.
Author Response

(The authors gave the same response as above.)

Round 2
Reviewer 1 Report
I thank the authors for clarifications and corrections.